# Activation of Neutrophils by Mucin–Vaterite Microparticles

**DOI:** 10.3390/ijms231810579

**Published:** 2022-09-13

**Authors:** Elena Mikhalchik, Liliya Yu. Basyreva, Sergey A. Gusev, Oleg M. Panasenko, Dmitry V. Klinov, Nikolay A. Barinov, Olga V. Morozova, Alexander P. Moscalets, Liliya N. Maltseva, Lyubov Yu. Filatova, Evgeniy A. Pronkin, Julia A. Bespyatykh, Nadezhda G. Balabushevich

**Affiliations:** 1Federal Research and Clinical Center of Physical-Chemical Medicine of Federal Medical Biological Agency, 119435 Moscow, Russia; 2Laboratory of Biomaterials, Sirius University of Science and Technology, 354340 Sochi, Russia; 3National Research Center of Epidemiology and Microbiology of N.F. Gamaleya, 123098 Moscow, Russia; 4Faculty of Chemistry, Lomonosov Moscow State University, 119991 Moscow, Russia; 5Expertise Department in Anti-Doping and Drug Control, Mendeleev University of Chemical Technology of Russia, 125047 Moscow, Russia

**Keywords:** mucin, vaterite, microparticles, chemiluminescence, cytokines, neutrophils, inflammation

## Abstract

Nano- and microparticles enter the body through the respiratory airways and the digestive system, or form as biominerals in the gall bladder, salivary glands, urinary bladder, kidney, or diabetic pancreas. Calcium, magnesium, and phosphate ions can precipitate from biological fluids in the presence of mucin as hybrid nanoparticles. Calcium carbonate nanocrystallites also trap mucin and are assembled into hybrid microparticles. Both mucin and calcium carbonate polymorphs (calcite, aragonite, and vaterite) are known to be components of such biominerals as gallstones which provoke inflammatory reactions. Our study was aimed at evaluation of neutrophil activation by hybrid vaterite–mucin microparticles (CCM). Vaterite microparticles (CC) and CCM were prepared under standard conditions. The diameter of CC and CCM was 3.3 ± 0.8 µm and 5.8 ± 0.7 µm, with ƺ-potentials of −1 ± 1 mV and −7 ± 1 mV, respectively. CC microparticles injured less than 2% of erythrocytes in 2 h at 1.5 mg mL^−1^, and no hemolysis was detected with CCM; this let us exclude direct damage of cellular membranes by microparticles. Activation of neutrophils was analyzed by luminol- and lucigenin-dependent chemiluminescence (Lum-CL and Luc-CL), by cytokine gene expression (IL-6, IL-8, IL-10) and release (IL-1β, IL-6, IL-8, IL-10, TNF-α), and by light microscopy of stained smears. There was a 10-fold and higher increase in the amplitude of Lum-CL and Luc-CL after stimulation of neutrophils with CCM relative to CC. Adsorption of mucin onto prefabricated CC microparticles also contributed to activation of neutrophil CL, unlike mucin adsorption onto yeast cell walls (zymosan); adsorbed mucin partially suppressed zymosan-stimulated production of oxidants by neutrophils. Preliminary treatment of CCM with 0.1–10 mM NaOCl decreased subsequent activation of Lum-CL and Luc-CL of neutrophils depending on the used NaOCl concentration, presumably because of the surface mucin oxidation. Based on the results of ELISA, incubation of neutrophils with CCM downregulated IL-6 production but upregulated that of IL-8. IL-6 and IL-8 gene expression in neutrophils was not affected by CC or CCM according to RT^2^-PCR data, which means that post-translational regulation was involved. Light microscopy revealed adhesion of CC and CCM microparticles onto the neutrophils; CCM increased neutrophil aggregation with a tendency to form neutrophil extracellular traps (NETs). We came to the conclusion that the main features of neutrophil reaction to mucin–vaterite hybrid microparticles are increased oxidant production, cell aggregation, and NET-like structure formation, but without significant cytokine release (except for IL-8). This effect of mucin is not anion-specific since particles of powdered kidney stone (mainly calcium oxalate) in the present study or calcium phosphate nanowires in our previous report also activated Lum-CL and Luc-CL response of neutrophils after mucin sorption.

## 1. Introduction

Calcium carbonate is one of the most common compounds in nature, and its polymorphs—calcite; aragonite; and especially vaterite, which is known for its high porosity—are becoming the most popular templates for drug encapsulation [1]. Nevertheless, crystallization and aggregation of calcium carbonate in some tissues such as the gallbladder, kidney, and bladder can contribute to the formation of stones [2]; for example, all CaCO_3_ polymorphs have been found in gallstones [3,4]. Vaterite is thermodynamically unstable and easily transforms into calcite or aragonite [5]. A number of proteins from bodily fluids are included in protein–mineral complexes during the course of biomineralization [6], and vaterite can be stabilized in the presence of biopolymers, amino acids, and peptides [7]. The major macromolecular components of mucus/mucin promotes aggregation of calcium salts in vivo [8,9,10] and was identified as a component of gallstones [9]. Mucin is considered as a good candidate for assembling microparticles due to its mucoadhesive properties [11], and its loading into the crystals via co-synthesis was twice as effective as via adsorption onto preformed crystals [12]. Mucins are highly glycosylated amphiphilic glycoproteins with molecular weights up to 20 MDa which comprise both membrane-bound and secreted subfamilies. A high proportion of their oligosaccharide chains terminates in negatively charged sialic acids [13]. We hypothesized that mucin could influence the reaction of the immune system to gallstones and others via effects on neutrophil activation. Reactive oxygen species (ROS) and hypochlorous acid generated by activated neutrophils can induce lipid peroxidation in host tissues and a cascade of pro-inflammatory mediators. Mineral–organic particles can more strongly induce neutrophil extracellular traps (NETs) formation in a ROS-dependent manner, especially microparticles of 1–2 µm in diameter, compared to nanoparticles [14]. By use of chemiluminescence, superoxide and degranulation assay, all three CaCO_3_ polymorphs were shown to cause rapid activation of neutrophils, with the potency of the crystals as aragonite > vaterite > calcite [15], but the role of mucin remained unclear. Our study was aimed at evaluation of neutrophil activation by hybrid mucin–vaterite microparticles. Vaterite microparticles were fabricated without mucin (CC) and with mucin (CCM) as described earlier [12,16]. Mucin from porcine stomach was used in CCM fabrication and further experiments. Normal human neutrophils were activated in vitro, and superoxide O2^·^ and HOCl production by the cells were assayed by lucigenin-chemiluminescence (Luc-CL) and by luminol-chemiluminescence (Lum-CL), respectively. To exclude damaging effects of the microparticles on cellular membranes, a hemolysis test [17,18] was used. Finally, a sample of human kidney stone was powdered and treated with mucin to control effects of its mucin coating on neutrophils.

## 2. Results

### 2.1. Characteristics of CC and CCM Microparticles

Incorporation of mucin into vaterite microparticles markedly altered their morphology and physical–chemical properties. According to SEM (Figure 1a,b), microparticles of CCM exceeded CC in diameter (5.8 ± 0.7 µm vs. 3.3 ± 0.8 µm), with nanocrystallites of smaller size (47 nm vs. 109 nm) and smoother surface. As shown by the BET method, the surface area of CCM was considerably superior to that of CC (37.6 m^2^ g^−1^ и 4.3 m^2^ g^−1^). Mucin incorporation into CCM amounted to 1.3 ± 0.3% *w*/*w*. Based on DLS data, CCM microparticles were more charged relative to CC (−7 ± 1 mV vs. −1 ± 1 mV), obviously because of glycoprotein coating.

### 2.2. Hemolytic Activity of CC and CCM Microparticles

The hemolysis assay is widely used to detect cell membrane destruction by microparticles. Hemolysis was evaluated by release of hemoglobin after incubation of human erythrocytes with microparticles for 2 h at 37 °C (Figure 1c). Even at a maximal ratio of particles to erythrocytes (5:1, corresponding to 1.67 mg mL^−1^), hemolysis by CC did not exceed 2%. Hemolytic activity of hybrid microparticles CCM was close to spontaneous values.

### 2.3. Neutrophil Chemiluminescence

Incorporation of mucin significantly increased the ability of CCM to activate both Luc-CL and Lum-CL of neutrophils compared to CC (Figure 2a,b), in a dose-dependent manner (Figure 2c,d).

The neutrophil membranes were not injured by contact with microparticles: after incubation of neutrophils with 5 mg mL^−1^ of CC or CCM at 37 °C for 30 min, no lysis was detected according to the data on extracellular LDH activity (Appendix A). Furthermore, within at least 30 min, there was no negative effect of particle concentration on the integral Luc-CL activity of neutrophils stimulated by CC or CCM followed by PMA (Appendix A), which was consistent with data of the LDH test and also confirmed cell viability.

When SOD was injected into a cuvette when the peak values of Luc-CL were reached, the signal dropped (for curves, see Appendix A), indicating a key role of superoxide anion in Luc-CL generation [19]. In the presence of 15 Un mL^−1^ of SOD, Luc-CL declined to 38 ± 2% compared with control values, and further addition of SOD up to 30 Un mL^−1^ did not affect CL intensity (Figure 2e). Hence, more than 50% of superoxide anion was generated extracellularly.

Lum-CL was inhibited by catalase, which decomposed hydrogen peroxide. As is known, in the presence of hydrogen peroxide and chloride ions, neutrophil MPO generates HOCl, which oxidizes luminol [20]. Catalase (10,000 Un mL^−1^) induced a drop of more than 50% in Lum-CL intensity (Figure 2f; for curves, see Appendix A).

The difference observed between CC and CCM in their neutrophil activating effects was surprising for us. One could expect that mucin would downregulate neutrophil response, as shown for bacteria [21] and artificial polymers [22]. In our study, we used mucin from porcine stomach, and to exclude eventual influence of its structural features, we compared activation of neutrophils with CC, CCM, and CC and zymosan coated with mucin by its adsorption (CC-M and Zym-M). In parallel, supernatants of particle suspensions were also analyzed under the same conditions (Table 1).

Mucin sorption by zymosan (0.31 mg g^−1^) was comparable to that of vaterite (0.23 mg g^−1^) and insignificantly changed zymosan ζ-potential (from −15.9 ± 1.8 V to −16.2 ± 0.8 V). Nevertheless, mucin sorption decreased neutrophil Lum-CL and Luc-CL response relative to zymosan, unlike mucin sorption by vaterite (Table 1). Hybrid vaterite microparticles (CCM) were more active than vaterite coated with mucin (CC-M). All studied vaterite suspensions activated neutrophils to a much greater extent compared to their supernatant fluids; hence, the eventual contribution of calcium ions and unbound mucin could be neglected.

The results of our previous investigations demonstrated that mucin was predominately located on the edges of the crystals and partially penetrated inside the crystal pores, and the presence of sialic acids in mucin most likely improved the loading and retention of mucin in crystals. This effect can be explained by the interaction of the acids with Ca^2+^ of the crystals. The loading efficiency of the desialylated mucin was found to be 33% and 36% lower compared to that for the loading of commercial mucin using adsorption and co-synthesis, respectively [12]. There was no linear dependence between mucin concentration in solution used for CC coating by adsorption and neutrophil CL response to these mucin-coated CC-M microparticles (Figure 3b). Thus, the decrease in loading efficiency in experiments with adsorption of desialylated mucin by 30% should not significantly reduce activation of neutrophils by CC-MD compared to CC-M. Nevertheless, vaterite microparticles coated by desialylated mucin (CC-MD) were significantly less active compared to CC-M and CCM (Figure 3a).

Since all experiments were carried out in the presence of 2% autologous serum, which was closer to natural bodily fluid, opsonization of microparticles with serum proteins could influence their activity towards neutrophils. The effects of opsonization of CC and CCM microparticles were analyzed (Figure 3c). No increase in CL intensity was detected after opsonization; in fact, opsonization reduced the amplitude of Lum-CL neutrophil response.

Lum-CL-response of neutrophils depends on MPO-catalyzed generation of highly reactive hypochlorous acid. It is known that mucin can be oxidized with 100–160 mM NaOCl, which destroys its protein core within a few minutes [23]. We incubated CCM microparticles in NaOCl solutions of various concentrations (up to 10 mM), and after thorough washing with NaCl, they were added to neutrophils. According to Luc-CL (Figure 3d) and Lum-CL (Figure 3e) amplitude of neutrophil activation response, the activating effect of CCM dropped depending on the NaOCl concentration used for their treatment.

### 2.4. Cytokine and Myeloperoxidase Release by Neutrophils

As shown by ELISA assay, incubation of neutrophils with CC or CCM for 1 h did not induce release of cytokines IL-1β, IL-6, or IL-10 or TNF-α compared to control (without particles), unlike suspension of *E. coli* (Table 2). Rather, CCM downregulated IL-6 release but upregulated IL-8 generation by neutrophils compared to control. No direct sorption of the studied cytokines by CC or CCM particles was detected. Since no significant changes in IL-6 and IL-8 expression were registered by RT^2^-PCR (Appendix A), ELISA-detected changes in IL-6 and IL-8 release induced by CCM resulted from posttranslational regulation of their synthesis.

There was a marked decrease in extracellular MPO concentration after incubation of neutrophils with CCM relative to control samples and CC, while CC did not affect MPO exocytosis.

MPO is a cationic protein of 150 kDa with pI 9.2 which favors its binding with negatively charged mucin (at pH 7.4). In our previous study, the adsorption of positively charged chymotrypsin onto hybrid vaterite–mucin microparticles was studied [24], and the activity of the bound chymotrypsin was evaluated as being the same as that of the free enzyme in solution. Thus, binding of the MPO which was released with stimulation by CCM microparticles could result in a drop of its concentration in extracellular buffer solution assayed for MPO by ELISA. At the same time, while adsorbed by CCM, MPO could participate in Lum-CL generation.

The ELISA assay of binding of commercial MPO by CC and CCM has shown that in the absence of blood serum proteins, MPO (120 ng mL^−1^) was totally adsorbed both by CC and CCM microparticles (5 mg mL^−1^), while treatment of CC with human serum albumin (HSA) blocked MPO adsorption (Appendix A). All experiments with neutrophils were carried out in a medium with 2% serum (~1.2 mg mL^−1^ HSA). Apparently, HSA prevented binding of MPO with CC, but not with CCM microparticles.

Evaluation of extracellular MPO activity showed a significant increase (~threefold) after incubation of neutrophils with *E. coli*, but not with CC or CCM (Appendix A).

### 2.5. Light Microscopy

Light microscopy of smears prepared after incubation of neutrophils with CC or CCM and control neutrophils (NaCl) showed that microparticles adhere to neutrophils (Figure 4a,b) and to their aggregates (Figure 4e) and are captured by NET-like structures (Figure 4f,g), as indicated by arrows. In the preliminary experiments, we compared staining of the smears using Hoechst dye (which is specific to DNA) and Romanowsky dye; both stains gave similar results (Appendix A).

We suppose that CCM microparticles adhere to neutrophil membranes and even cause their invagination, but their phagocytosis was not decisively demonstrated (Appendix A) compared to zymosan (Appendix A). Neutrophils incubated with CCM were more prone to aggregation (Figure 4d,e) and forming NET-like structures (Figure 4f,g).

### 2.6. Analysis of Human Kidney Stones

The suspension of kidney stones (St) from a patient with urolithiasis was incubated with mucin solution and washed with 0.15 M NaCl; this treated-with-mucin sample (St-M) and St were used for neutrophil activation. The suspensions consisted of microparticles of 6 ± 5 µm, according to SEM. St-M suspension showed a much stronger activating effect compared to St, according to data of Luc-CL (Table 3). Antibiotic polymyxin B sulfate (PMB), which is a well-known agent for removal of endotoxins, was added to parallel probes to test eventual effects of lipopolysaccharides. The same test was performed with CC and CCM suspensions. Lipopolysaccharide O111:B4 (50 µg mL^−1^) was used as a positive control.

Obviously, PMB did not affect neutrophil response to the particles, unlike O111:B4 solution, and did not influence the effects of mucin.

## 3. Discussion

Polymorphonuclear neutrophils play a central role in host defense and inflammation induced by infection and by both biomineral and non-biological particles. The binding of particles to the plasma membrane of neutrophils results in the production of reactive oxygen species (ROS) by the cells via activation of NADPH oxidase (respiratory burst); reaction of hydrogen peroxide with chloride catalyzed by neutrophil myeloperoxidase, with HOCl production; neutrophil degranulation; and cytokine secretion [15,25]. Activated neutrophils can externalize their chromatin decorated with granular proteins in the process of NET formation [26], driven mainly by ROS [14,27]. Biomineral nanoparticles which form spontaneously in human bodily fluids may become pro-inflammatory and contribute to pathological processes once they aggregate and form larger mineral and protein–mineral particles [14,28]. For example, NET formation induced by such nano- and micro-objects is considered to be a potent factor provoking formation of gallstones [29], pancreatic stones [26], and salivary stones [30], while excessive ROS and HOCl can cause damage to lipids, proteins, and DNA [31].

The interaction of neutrophils and other phagocytes with biomineral particles depends on the particle size, shape, hydrophilicity, surface charge, surface coating, and roughness [32]. Elongated CaCO_3_ microparticles are internalized by HeLa cells with a higher frequency than spherical particles [33], and submicron needle-shaped aragonite particles trigger pro-inflammatory response of THP-1 macrophages, unlike cuboidal-shaped calcite particles [34]. As shown with polystyrene particles, it is the local particle shape, measured by tangent angles, at the point of initial contact which dictates whether macrophages initiate phagocytosis or simply spread on particles [35]. The effect of particle size remains disputable. The ability of CaCO_3_ nanoparticles to stimulate the production of ROS leading to oxidative stress, inflammation, genotoxicity, metabolic changes, and potentially carcinogenesis was recently reviewed [36]. As for the microparticles, neutrophils and other phagocytes specifically internalize particulate targets (typically > 0.5 µm in diameter) by actin-dependent phagocytosis processes via diverse mechanisms [37]. The pro-inflammatory potential of hydroxyapatite crystals towards macrophages in vitro decreased with an increase in particle diameter from 1µm to 10 µm and in surface pore size [38]. Crystalline calcium phosphate-based mineral–organic particles of a few micrometers in diameter elicit neutrophil pro-inflammatory reactions and NET formation, unlike particles of small sizes (<100 nm) [14]. Particle hydrophilicity, surface charge, and roughness depend on their coating with other molecules, especially in bodily fluids.

Orally administered calcium carbonate nanoparticles can rapidly interact with components of biological fluids, with formation of corona composed of proteins [39]. Endogenous nanoparticles in blood trap plasma proteins [28,40] and co-precipitate with mucin in the gastrointestinal tract [3,10]. As was shown in the model studies, mucin is easily adsorbed by calcium carbonate microparticles (vaterite) and is also incorporated into vaterite by co-precipitation [12]. Nevertheless, the effects of mucin—both incorporated into biomineral particles or coating them—is still unclear; in the present study, we investigated effects of vaterite–mucin microparticles.

We activated neutrophils in vitro with vaterite (CC), vaterite–mucin hybrid (CCM), and mucin-coated vaterite (CC-M) microparticles. The porcine gastric mucin used in the present study (as in our previous studies [11,12,16,41,42,43]) has been shown to be structurally related to human gastric mucin [44] and is, therefore, a decent substitute for human gastric mucin because of its high availability and reduced number of ethical issues required for the research.

Mucin increased the negative charge of microparticles as follows: CC < CCM < CC-M, where CC-M charge exceeded that of CCM by 1.4 times [12]. None of the samples demonstrated marked hemolytic activity, which is consistent with available data of other researchers for CaCO_3_ microparticles [17]. This allows us to exclude direct membrane damage of blood cells by the microparticles. Chemiluminescent response of activated neutrophils dependent on superoxide anion production (Luc-Cl) [19] and hypochlorous acid production (Lum-Cl) [20] induced by microparticles increased as CC < CC-M < CCM, unlike their surface charge. Neither dissolved mucin nor supernatants of the microparticle suspensions activated neutrophils to the same degree as microparticles. All these results indicated that calcium or mucin elution from the microparticles or their surface charge were not the main factors in neutrophil activation.

Production of ROS by neutrophils could be influenced by opsonization of microparticles with blood serum immunoglobulins [15,39], but there was no serum-derived increase in chemiluminescent in our study. The pro-inflammatory effect of mucin incorporated into or adsorbed onto vaterite microparticles, detected as increased oxidant production, is consistent with data of our previous study concerning activation of neutrophils with polydisperse calcium phosphate nanowires coated with mucin [43]. A pilot experiment with a kidney stone also showed that mucin adsorption enhanced Lum-CL and Luc-CL response of neutrophils induced by the powdered sample. Since calcium oxalate is the major component of kidney stones [45], it becomes obvious that the neutrophil-activating role of mucin in mucin-coated or hybrid biomineral particles is not anion-specific. These results contradict the data of experiments with bacteria. Mucin was shown to protect Gram-negative bacteria from neutrophil attack, preventing complement deposition on bacteria surface [41,42], by direct suppression of reactive oxygen species (ROS) production [21]. In the present study, we demonstrated that mucin sorption by zymosan (yeast cell walls) downregulated CL response of neutrophils to this activator. Hence, the difference between CC and zymosan in mucin effects did not result from eventual mucin structural features [46].

Surface mucin can facilitate activation of neutrophils via morphologic change of vaterite surface from rough to smooth, as shown by SEM. Mucins are decorated with epitopes whose ligands can be found on neutrophils. Mucin sialylated and fucosylated O-glycans can be recognized by neutrophil sialic acid-binding immunoglobulin-like lectins (Siglec receptors) [47] and L-selectin [48]. Neutrophil Siglecs suppress neutrophil activation [49,50], but in our study, desialylation of mucin did not enhance ROS production by neutrophils stimulated by CC-MD relative to CC-M. Moreover, the amplitude of chemiluminescent response to CC-MD was lower than that to CC-M and CCM, so the direct role of Siglecs in neutrophil activation by vaterite–mucin is not obvious.

Such oligosaccharide determinants of mucin as sialylLe^x^ (NeuAcα2→ 3Galβ1→ 4[Fucα1→ 3]GlcNac) are recognized by neutrophil L-selectin [51]. Both density and clustering of these carbohydrate ligands affect neutrophil contact with microparticles [52]. Thus, highly porous vaterite–mucin microspheres CCM composed of smaller nano-crystallites could provide preferential conditions for neutrophil activation compared to CC and even CC-M. Oxidation of mucin in mucin–vaterite microparticles with 0.2–10 mM NaOCl resulted in reduction of their neutrophil-activating capacity, presumably because of protein core oxidative damage [23]. This result proves that it is mucin in hybrid particles that participates in activation of neutrophils. Neutrophils that accumulate in the interstitial fluid of inflamed tissues have been reported to produce HOCl at concentrations of up to 25–50 mM/h [53]. Considering that pKa of HOCl is close to 7.5 [54] and that, at physiological pH values, about half of the acid is in the molecular form and the rest of the acid in the form of anions, the HOCl/OCl^-^ mixture is usually present in the cell media. Thus, our result also shows that mucin covering the hybrid microparticles could be oxidized by neutrophil-derived HOCl/OCl^-^, at least partially. One could assume that mucin–vaterite microparticle-induced production of oxidants by neutrophils is aimed at surface mucin damage to prevent further precipitation of calcium carbonate (or calcium phosphate) by this glycoprotein.

The neutrophil activation by CCM was characterized not only by an increase in ROS and HOCl generation, but also by cell aggregation and formation of NET-like structures. Unlike *E. coli*, CCM did not stimulate any significant cytokine release except for a slight increase in IL-8 production and decrease in IL-6 secretion, both due to posttranslational regulation. Our results are consistent with data of other researchers concerning activation of human dendritic cells by intestinal mucin [55]. Despite the ability of intestinal mucin to induce IL-8, there was no effect on production of other pro-inflammatory cytokines (TNF-α, IL-6, and IL-23) or the anti-inflammatory IL-10.

NETosis is considered as a regulated process [56]. NADPH oxidase function is crucial for the formation of NETs [57], and inhibition of peptydilarginine deiminase type 4 or abrogation of reactive oxygen species inhibits NET-dependent gallstone formation in vivo [29]. MPO-derived reactive halogen species also participate in NETosis [57]. Thus, activation of ROS production registered by Luc-CL and RHS registered by Lum-CL induced by CCM could be considered as potential triggers for NET-like structure formation.

In summary, the regulation of neutrophil activity is aimed at triggering oxidant production and specific autocrine and paracrine signaling mechanisms via IL-8 secretion, but further work is required to determine how this scenario is realized, especially in patients with urolithiasis or gallstone disease. Indeed, there is an interplay between mucus, microbiome, and immunity in such patients, dependent also on the patient’s age. Age is considered as an independent risk factor for cholesterol gallstone formation [58,59]. Decreases in butyrate, lactate, acetate/propionate, and methane producers; mucin-degrading bacteria; and increases in lipopolysaccharide positive bacteria were registered in patients with cholesterol-rich gallstones, triggering chronic and acute inflammation [59,60]. Inflammation-induced biliary mucin production promotes cholesterol crystallization [61], and mucin exposed on the gallstones can enhance inflammation via reaction with granulocytes.

## 4. Materials and Methods

### 4.1. Reagents

CaCl_2_ (≥93.0%), Na_2_CO_3_ (≥99.0%), mucin from porcine stomach (type III, with 0.5–1.5% bound sialic acid), NaOCl, zymosan, Histopaque (1.077, 1.119), Krebs–Ringer solution, luminol, lucigenin, SOD (3000 U mg^−1^), catalase (24,640 U mg^−1^, solid), and polymyxin B sulfate were purchased from Sigma-Aldrich (St. Louis, MO, USA). Hemoglobin standard solution (68 g L^−1^) was manufactured by OOO Agat, Moscow, Russia. ELISA kits for cytokines IL-1b, IL-6, and IL-8 and TNF-alpha detection (Cytokine, Sankt-Petersburg, Russia); MPO ELISA kit (Cloud-Clone Corp, Katy, TX, USA); OT-PCR kits “Proba-NK” (DNA-technology, Moscow, Russia); and “Reverta-L” (AmpliSens, Moscow, Russia) were purchased from manufacturers. Liquid culture of *E. coli* Mg1655 was kindly provided by Dr. O.V. Pobeguts (laboratory of proteome analysis, Federal Research Clinical Center of Physical-Chemical Medicine, Moscow, Russia).

### 4.2. Analytical Determination of Mucin

The concentration of the glycoprotein mucin in solution was determined using analytical size exclusion chromatography in the Biofox 17 SEC 8 × 300 mm column (Bio-Works, Uppsala, Sweden), utilizing the Smartline chromatographic system (Knauer, Berlin, Germany) in a solution of 0.15 M NaCl [12]. Preliminarily, the column was calibrated by solutions of purified mucin with different concentrations (0.01–1.00 mg mL^−1^). A total of 0.02 mL of the mucin solution was used for the chromatography analysis at an elution rate of 0.5 mL min^−1^. Absorbance of the eluted solutions was measured using a UV detector at wavelengths 214 and 260 nm.

### 4.3. Desialylation of Mucin

A 50 mg amount of mucin was dissolved in 50 mL of 0.01 M HCl and incubated for 3 h at 80 °C [11]. The solution was chromatographed at the column filled with Sephadex G-200 (dimensions 2.5 × 35 cm), using the chromatographic system Bio-Logic LP (Bio-Rad, Hercules, CA, USA) in a solution of ammonia (pH of 9.0) [41]. The elution rate was 0.5 mL min^−1^; collection time for one fraction of eluted solution was 12 min. Absorbance was determined in the obtained fractions at wavelengths of 214 and 260 nm. The fractions containing mucin, as identified by absorbance and specific determination by the Schiff method, were combined and freeze-dried.

The sialic acid content, determined by Hess’s method [62] using the calibration curve for N-acetylneuraminic acid, was found to be 2.30 ± 0.10 and 0.41 ± 0.05% for commercial and desialylated mucin (MD), respectively.

### 4.4. Fabrication of Vaterite Microparticles

Vaterite microparticles (CC) were synthesized as described earlier [16]. The mixture of 9 mL of 0.05 M Tris buffer (pH 7.0) with 0.3 mL 1M CaCl_2_ (pH 7.0) and 3 mL 0.1 M Na_2_CO_3_ was stirred at RT, and the formed crystals were washed twice with pure water and lyophilized. For fabrication of hybrid microparticles with mucin, 3 mL of 1 M CaCl_2_ containing 8.3 mg mL^−1^ of mucin in 0.05 M Tris buffer (pH 7.0) was stirred for 10 min, followed by addition of 3 mL of 1 M Na_2_CO_3_ water solution. The precipitate was separated by centrifugation for 2 min at 1000 × *g*, washed twice with double-distilled water, and lyophilized. To calculate the mucin content of CCM microparticles, its concentration in the supernatant and the washing solutions was assayed as described above.

### 4.5. Mucin Adsorption onto CC and Zymosan Particles

A suspension of CC or zymosan (Zym) (10 mg mL^−1^) in 0.15 M NaCl was mixed with an equal volume of mucin solution (10 mg mL^−1^) and incubated for 30 min at 37 °C under periodic shaking. The precipitate was separated by centrifugation for 10 min at 1000 × *g*, washed twice with 0.15 M NaCl, and resuspended to 10 mg mL^−1^ (CC-M or Zym-M).

### 4.6. Opsonisation

Equal volumes of the CC or CCM suspension (10 mg mL^−1^) in 0.15 M NaCl and normal human serum were thoroughly mixed and incubated at 37 °C for 30 min with periodic shaking. Then, the mixtures were centrifuged (10 min at 1000 × *g*), washed twice with 0.15 M NaCl, and resuspended to 10 mg mL^−1^ (CC-ops or CCM-ops).

### 4.7. Treatment of Microparticles with NaOCl

A suspension of CC or CCM microparticles (10 mg mL^−1^) in 0.15 M NaCl was mixed with an equal volume of NaOCl water solution—so that its final concentration was 0.1–10 mM, in accordance with the study design—and incubated at 37 °C for 30 min. Then, the microparticles were precipitated with centrifugation, washed thrice with 0.15 M NaCl, and resuspended to 10 mg mL^−1^.

### 4.8. Scanning Electron Microscopy (SEM)

Immediately before sample deposition, silicon wafers were treated in plasma cleaner Electronic Diener (Plasma Surface Technology, Regensburg, Germany). The CC particles were then deposited onto them and characterized using a Zeiss Merlin microscope equipped with GEMINI II Electron Optics (Zeiss, München, Germany). SEM parameters were accelerating voltage (1–3 kV) and probe current (30–80 pA).

### 4.9. ζ- Potential Measurement

The ζ-potentials of microparticles and mucin was measured using Zetasizer (Nano ZS, Malvern, UK) and estimated using the Smoluchowski equation. Mucin ζ-potential measured in water was −6 ± 1 мB.

### 4.10. Nitrogen Adsorption–Desorption by the Brunauer–Emmett–Teller (BET) Method

The porosity of vaterite microspheres was studied by low-temperature nitrogen adsorption–desorption by the Brunauer–Emmett–Teller (BET) method on an ASAP-2000 setup (Micromeritics, Norcross, GA, USA). The samples were preliminarily subjected to vacuum treatment at room temperature to a residual pressure of 2 × 10^−6^ bar. The isotherms of N_2_ sorption were recorded at −196 °C as the dependences of the volume of sorbed nitrogen (cm^3^/g) on the relative pressure p/p^0^, where p^0^ is the pressure of saturated nitrogen vapours at −196 °C. The porous characteristics of the samples were calculated using a standard software package. The specific surface area was estimated by the BET method; the pore size was calculated using the Barrett–Joyner–Halenda (BJH) method.

### 4.11. Blood Cell Isolation

Human erythrocytes and neutrophils were isolated from the normal blood of 4 volunteers aged 35–60 years without inflammatory diseases, on the basis of their informed consent and agreement. Erythrocytes were sedimented by centrifugation of each blood sample collected (at 400× *g*, with EDTA as anticoagulant) and then washed thrice with 0.15 M NaCl. Another blood volume was layered over the double gradient of Histopaque 1.077/1.119 g L^−1^, and after centrifugation for 45 min, neutrophils were collected and washed with Krebs–Ringer solution. Cell concentration was assayed by direct counting using a Goryaev chamber, with a neutrophil purity of >95%.

### 4.12. Hemolytic Activity Assay

A 40 µL amount of of CC or CCM suspension (10 mg mL^−1^) was diluted with 180 µL of 0.15 M NaCl and mixed with 40 µL of erythrocytes (5–10 × 10^9^ cell mL^−1^). The mixture was incubated for 2 h at 37 °C. Then erythrocytes and particles were separated by centrifugation, and supernatants were collected for further hemoglobin (Hb) assay by absorbance at 540 nm using a standard Hb solution. The total hemolysis was induced by distilled H_2_O; to assay spontaneous hemolysis, 0.15 M NaCl was added to erythrocytes instead of particles. The results are expressed in %: HA = (Hb_x_ − Hb_sp_)/(Hb_t_ − Hb_sp_) × 100%, where HA—hemolytic activity (%); Hb_x_, Hb_t_, Hb_sp_—hemoglobin concentration (mg mL^−1^) in supernatants of the probes with particles, H_2_O or NaCl, respectively.

### 4.13. Chemiluminescent Assay (CL)

CL was measured using the luminometer Lum1200 (DiSoft, Moscow, Russia) in 0.5 mL of Krebs–Ringer solution (pH 7.4) with 0.2 mM luminol (Lum-CL) or lucigenin (Luc-CL), 2% of autologous blood serum, neutrophils (0.5–0.7 × 10^6^ cells mL^−1^), and particles (1 mg mL^−1^). Spontaneous CL was measured before the addition of particles; then, the sample was added and CL was registered until maximum values were reached. the CL amplitude (V) was calculated as the difference between peak and spontaneous values. If necessary, superoxide dismutase (SOD) or catalase (Cat) was added just when the peak values were achieved.

### 4.14. Cytokine RNA Expression

Cytokine RNA were detected by means of reverse transcription with subsequent real-time PCR (RT^2^-PCR) with fluorescent hydrolysis probes. Total nucleic acids were isolated from 50 µL of control intact and experimental neutrophils after their incubation with CC or CCM for 1 h at 37 °C, by using “Proba-NK” kit (DNA-technology, Moscow, Russia). Control neutrophils were incubated with 0.15 M NaCl. Then, the reverse transcription was performed using “Reverta-L” kit (AmpliSens, Moscow, Russia). To detect mRNA of human interleukines with specific primers and probes, RT^2^-PCR was performed as previously described [63].

### 4.15. Cytokine and Myeloperoxidase ELISA Assay

Neutrophils (2 × 10^6^ cells mL^−1^) were incubated with 1 mg mL^−1^ of particles or with *E. coli* Mg1655 (2.5 × 10^8^ CFU mL^−1^) in Krebs–Ringer solution with 2% blood serum for 1 h at 37 °C. Supernatants after centrifugation for 20 min at 900× *g* were collected and stored at −60 °C until analysis. Cytokines (IL-1β, IL-6, IL-8, and IL-10) and myeloperoxidase (MPO) were assayed using ELISA kits according to manufacturer instructions.

### 4.16. Light Microscopy

The standard smears of neutrophils incubated with particles were prepared in autologous blood plasma without any cell centrifugation, fixed with methanol, and stained according to the Romanovskii–Giemsa technique; then, they were analyzed using a light Motic BA223 microscope (Motic, Kowloon Bay, Hong Kong) equipped with a 3CCD KYF32 digital camera. Image processing was performed with a MECOS-C image analysis system (MECOS, Russia) in semi-automatic mode (400×).

### 4.17. Kidney Stone Collection and Processing

Stones of the right kidney were obtained from a patient diagnosed with urolithiasis, on the basis of their informed consent and agreement. Stone samples were collected after surgical procedure for removal (percutaneous nephrolithotomy), with a portion of the sample sent for clinical analysis of composition. Remaining stone samples were rinsed with sterile PBS to remove potential host bacteria contamination. Stones were fractionated with a sterile mortar and scalpel and stored at −80 °C before future use.

Stones were powdered in 0.15 M NaCl with Potter S homogenizer (Sartorius, Göttingen, Germany) until a homogeneous suspension (10 mg mL^−1^) was achieved. The processing of further samples with mucin solution (St-M) or 0.15 M NaCl (St) was performed as described above (“Mucin adsorption” section). The precipitation was washed twice with 20-fold excess 0.15 M NaCl and resuspended to 10 mg mL^−1^ or subsequent activation of isolated human normal blood neutrophils, according to the procedure described above (“Chemiluminescent assay” section).

### 4.18. Statistics

Statistical significance levels were calculated with the Student’s *t*-test using Statistics 12 (StatSoft). *p* values < 0.05 were assumed to be significant. Data are represented as mean value ± standard deviation.

## 5. Conclusions

The main features of neutrophil reaction to mucin–vaterite hybrid microparticles are increased oxidant production, cell aggregation, and NET-like structure formation, but without significant cytokine release (except for IL-8).

## Figures and Tables

**Figure 1 ijms-23-10579-f001:**
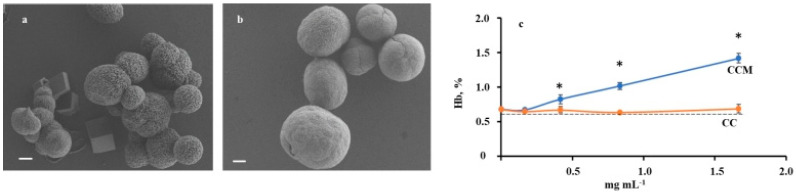
SEM of CC (**a**) and CCM (**b**) microparticles (scale bar 1 µm); effect of CC and CCM concentration on hemolysis assessed by Hb release (**c**). Erythrocytes (0.8 × 10^9^ cells mL^−1^) in 0.15 M NaCl were incubated with microparticles for 2 h at 37 °C. Dotted line corresponds to spontaneous erythrocytes lysis. All measurements were performed in triplicate. * *p* < 0.001 vs. CC.

**Figure 2 ijms-23-10579-f002:**
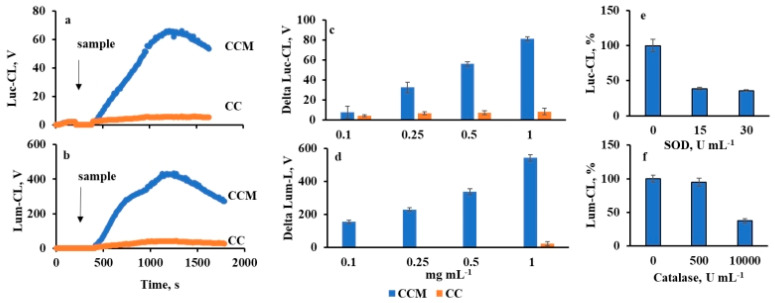
Neutrophil CL response activated by CC (in orange) and CCM (in blue): time-course of neutrophil Luc-CL (**a**) and Lum-CL (**b**) at particle concentration of 1 mg mL^−1^; dose-dependence of CC and CCM effects on neutrophil Luc-CL (**c**) and Lum-CL (**d**) amplitude; effect of SOD on CCM-induced Luc-CL (**e**) and effect of catalase on Lum-CL (**f**) peak values (%). Neutrophil concentration was 0.6 × 10^6^ cells mL^−1^. Each value was measured in triplicate.

**Figure 3 ijms-23-10579-f003:**
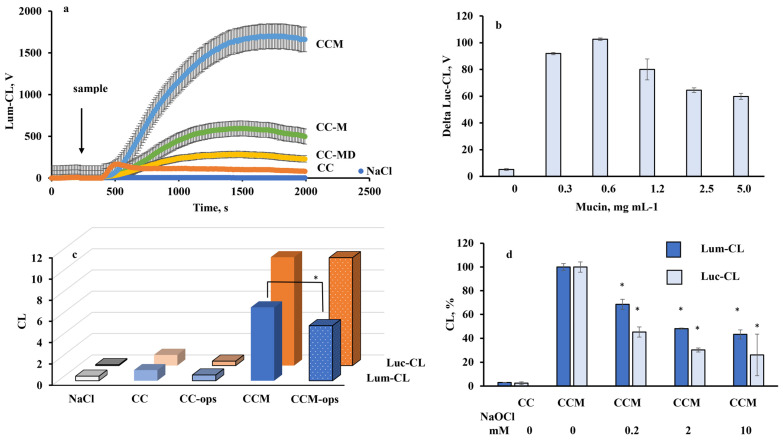
Chemiluminescent response of activated neutrophils: (**a**) Time-course of Lum-CL response to CC, CC-MD, CC-M (CC coated with mucin by adsorption), and CCM (prepared by CC co-precipitation with mucin). The arrow indicates the time-point of sample addition. Sample concentration was 1 mg mL^−1^; neutrophil concentration was 0.6 × 10^6^ mL^−1^. (**b**) Effects of concentration of mucin in solution for CC coating on Luc-CL amplitude. (**c**) Effects of opsonization of microparticles with autologous serum on Luc-CL and Lum-CL peak values normalized by response to CC microparticles (* *p* ˂ 0.05 vs. CCM). (**d**) Effects of concentration of NaOCl at incubation with CCM on subsequent activation of neutrophils assessed as Luc-CL and Lum-CL peak values. Data are represented as % to samples with untreated CCM (* *p* ˂ 0.05 vs. untreated CCM). Each value was measured in triplicate. Concentration of microparticles was 1 mg mL^−1^.

**Figure 4 ijms-23-10579-f004:**
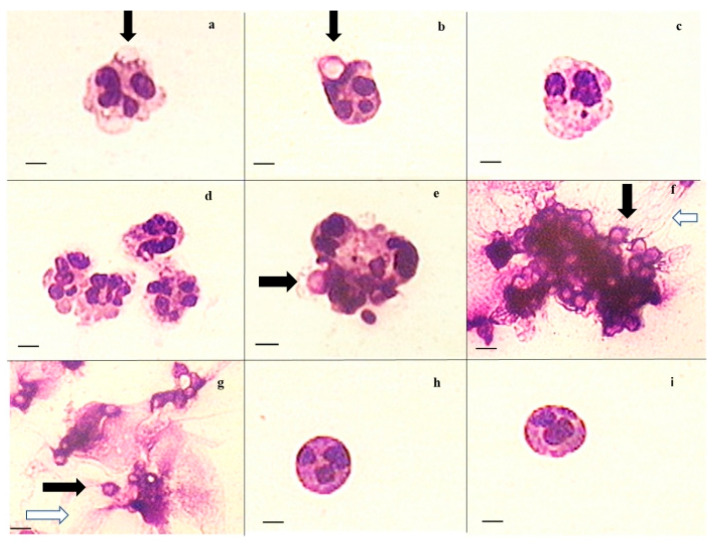
Typical details in light microscopy microphotographs of neutrophils incubated for 1 h with CC (**a**–**c**), CCM (**d**–**g**), and control with saline aliquot (**h**,**i**) (scale bars 10 µm). Black arrows indicate microparticles; white arrows indicate NET-like structures.

**Table 1 ijms-23-10579-t001:** Amplitude of Lum-CL and Luc-CL neutrophil response activated by microparticles and corresponding supernatant aliquots. Data of three independent experiments were normalized by the value of neutrophil response to CC microparticles and are represented in conventional units as average value and standard deviation. In each experiment, the values were measured in triplicate.

Samples	Lum-CL		Luc-CL	
	Suspension	Supernatant	Suspension	Supernatant
**CC**	1.0 ± 0.03	0.9 ± 0.01	1.0 ± 0.7	0.6 ± 0.3
**CCM**	44.7 ± 1.3 *	3.9 ± 0.12 *	75.5 ± 0.5 *	4.1 ± 0.4 *
**CC-M**	15.4 ± 0.4 *^#^	2.2 ± 0.07 *	36.7 ± 11.5 *^#^	3.0 ± 0.5 *
**Zym**	120 ± 0.9	n.d.	48.0 ± 5.4	n.d.
**Zym-M**	81.4 ± 2.6 **	n.d.	36.2 ± 4.0 **	n.d.
**Mucin 1 mg mL^−1^**	2.2 ± 0.05	0.2 ± 0.01	6.2 ± 0.6	7.8 ± 0.4
**NaCl**	0.2 ± 0.01	0.2 ± 0.01	0.4 ± 0.3	0.4 ± 0.3

* *p* ˂ 0.01 vs. CC. ** *p* ˂ 0.01 vs. Zym. ^#^
*p* ˂ 0.01 vs. CCM.

**Table 2 ijms-23-10579-t002:** Extracellular concentration of cytokines and myeloperoxidase after incubation of neutrophils with CC and CCM microparticles and bacteria *E. coli* for 1 h at 37 °C. All measurements were performed in triplicate.

Samples	Control (0.15 M NaCl)	CC	CCM	*E. coli*
**IL-1β, pg mL^−1^**	0.7 ± 0.2	0.3 ± 0.2	0.9 ± 0.5	**1.7 ± 0.5 ***
**IL-6, pg mL^−1^**	11.1 ± 3.0	6.9 ± 3.2	5.4 ± 0.7 *	**12.9 ± 5.5**
**IL-8, pg mL^−1^**	45.3 ± 5.5	51 ± 2.4	58 ± 5 *	**51 ± 5**
**IL-10, pg mL^−1^**	2.7 ± 0.7	2.1 ± 0.2	3.3 ± 1.5	**5.9 ± 3.2 ***
**TNF-α, pg mL^−1^**	1.4 ± 0.5	0.6 ± 0.3	2.4 ± 1.1	**6.1 ± 1.1 ***
**MPO, ng mL^−1^**	307 ± 78	313 ± 7	175 ± 41 *	n.d.

* *p* < 0.05 vs. control.

**Table 3 ijms-23-10579-t003:** Amplitude of Luc-CL response (V) of neutrophils stimulated with suspension of St or St-M suspension and CC or CCM particles in the presence of 20 µM polymyxin B sulfate (PMB) or without PMB (0.15 M NaCl was added). All measurements were performed in triplicate.

Samples	0.15 M NaCl	20 µM PMB
**St**	0.8 ± 0.4	1.0 ± 0.3
**St-M**	31.5 ± 0.8 *	37.3 ± 7.3 *
**O111:B4**	5.6 ± 1.3	0.5 ± 0.2 ^#^
**CC**	1.2 ± 0.1	0.9 ± 0.2
**CCM**	46.7 ± 3.7 **	60.4 ± 13.5
**NaCl**	0.7 ± 0.4	0.6 ± 0.2

* *p* < 0.001 vs. St. ** *p* < 0.001 vs. CC. ^#^
*p* < 0.001 vs. O111:B4, 0.15 M NaCl.

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
