# Peer review of "Activation of Neutrophils by Mucin–Vaterite Microparticles"

_ijms, 2022, doi:10.3390/ijms231810579_

Round 1
Reviewer 1 Report
Authors described that the main features of neutrophil reaction to mucin-vaterite hybrid microparticles (CCM) are increased oxidants production, cell aggregation, and NETs-like structure formation. It is very interesting to note that supporting neutrophil activation of mucin-vaterite hybrid microparticles in humans. However, these findings in the manuscript have some problems as follows.
Major revise
What is the pathogenesis of the formation of NETs-like structures after CCM activates neutrophils? The fact that NETs-like structures are caused by CCM should be discussed. Also, although a hemolysis assay has been performed on CCM, how is the cytotoxicity of CC-M or CCM to neutrophils?
Although the authors state that neutrophils caused NET-like structures upon exposure to CCM, it seems that only light microscopy would be insufficient to evaluate NETs like formation. Is it possible to evaluate decondensed chromatin on neutrophils?
In Figure 4, there is a difference in the number of particles phagocytosed by the neutrophils (Figure 4 a, f). It seems that the neutrophils which phagocytized more particles were simply collapsed by the CCM microparticles. Authors should be presented the data and discussion supporting their suggestion that CCM caused the formation of something like NETs should be clearly presented.
Why is E.coli not measured only for MPO of E.coli, which is supposed to be used as a positive control in Table 2. The results of the analysis of MPO of the positive control are important in evaluating MPO reduction in CCM.
Authors showed the CCM microparticles induced the increase of Lum-CL-response, which neutrophils depend on MPO-catalysed generation of highly reactive hypochlorous acid, in Figure 3a and Table 1. However, MPO concentration by CCM exposure was deceased in Table 2. Authors should discuss the discrepancy between Lum-CL-response and decrease of MPO.
Minor revise
Figures 2, 3, and Table 1 should list the number of N in each group.
The Legend of Figure 3 b is incorrect. Luc-CL is correct, not Lum-CL.
Reviewer 2 Report
In the article, "Activation of neutrophils by mucin-vaterite microparticles", the authors aimed at evaluating the activation of neutrophils by CCM. The study is well conducted and well presented with sufficient background and significance.
There are some minor points that I would like authors to address.
1. What was the age of the donors and were they suffering from any co-morbidities that could influence the neutrophil activation assays? I would encourage the authors to add this information in the Methods section.
2. What was the %purity of neutrophils obtained from the gradient centrifugation?
3. The study highlights the potential of mucin to activate the neutrophils in situations of gallstone presence in the body which could lead to serious tissue pathology. This issue is more problematic in the aged population with increased risk of gallstone formation and biomineralization. I would encourage the authors to highlight this aspect in their discussion section which would further highlight the significance of their findings.
Thanks
Round 2
Reviewer 1 Report
Authors answered all my questions and doubts as well as corrected the manuscript. In this regard, I suggest acceptance of the presented revised manuscript.